# Land-Greening Hotspot Changes in the Yangtze River Economic Belt during the Last Four Decades and Their Connections to Human Activities

**Liangsheng Zhang [1,2], Haijiang Luo [3] and Xuezhen Zhang [2,4,*]**

1    College of Geography and Environment, Shandong Normal University, Jinan 250358, China;
     2019020688@stu.sdnu.edu.cn
2    Key Laboratory of Land Surface Pattern and Simulation, Institute of Geographic Sciences and Natural
     Resources Research, CAS, Beijing 100101, China
3    China National Environmental Monitoring Centre, Beijing 100012, China; luohj@cnemc.cn
4    College of Resources and Environment, University of Chinese Academy of Sciences, Beijing 100049, China
*    Correspondence: xzzhang@igsnrr.ac.cn; Tel.: +86 134-8876-0873

**Abstract:** The spatial patterns of the normalized difference vegetation index (NDVI) changes in the Yangtze River Economic Belt (YREB) and their potential causes during the last four decades remain unclear. To clarify this issue, this study firstly depicts the spatial patterns of the NDVI changes using global inventory modelling and mapping studies (GIMMS) NDVI data and Moderate Resolution Imaging Spectroradiometer (MODIS) NDVI data. Secondly, the Mann–Kendall test, regression residual analysis and cluster analysis are used to diagnose the potential causes of the NDVI changes. The results show that the regional mean NDVI exhibited an uptrend from 1982 to 2019, which consists of two prominent uptrend periods, i.e., 1982–2003 and 2003–2019. There has been a shift of greening hotspots. The first prominent greening trend from 1982 to 2003 mainly occurred in the eastern agricultural area, while the second prominent greening uptrend from 2003 to 2019 mainly occurred at the junction of Chongqing, Guizhou and Yunnan. The greening trend and shift of greening hotspots were slightly caused by climate change, but mainly caused by human activities. The first greening trend was closely related to the agricultural progress, and the second greening trend was associated with the rapid economic development and implementation of ecology restoration policies.

**Keywords:** temporal and spatial variation of vegetation; human activities; residual analysis; cluster analysis

## 1. Introduction

As a natural link between soil, atmosphere and water, vegetation plays a fundamental role in the ecosystem by participating in energy exchange and the carbon cycle [1,2]. Vegetation change is usually considered as an indicator of environmental changes and human activities. Therefore, it is of great scientific significance to study the spatial–temporal characteristics of vegetation changes and their underlying mechanism, which is currently a research hot topic. In recent years, scholars have usually used remote sensing data to monitor vegetation change and have proposed many vegetation indices, such as the normalized difference vegetation index (NDVI), the ratio vegetation index (RVI), the difference vegetation index (DVI), etc. Among them, NDVI is the most widely used and widely recognized vegetation index [3–5]. Due to advances in remote sensing technology and improvements in datasets, long-term NDVI datasets have been widely used to monitor vegetation dynamics [6].

Vegetation change is influenced by climate, land-use change and other anthropogenic factors [7]. Many studies have shown that change in precipitation as well as temperature over a time period significantly impact NDVI variations [8]. Vegetation coverage is positively correlated with precipitation in most dry areas, whereas in humid areas, there was

a negative correlation between vegetation conditions and heavy rainfall [9,10]. In recent years, large-scale human economic and ecological activities have had a great impact on the environment [11]. Anthropogenic factors play a more important role in spatial–temporal change of vegetation [12]. For example, to improve the quality of the ecological environment, the Chinese government has implemented many vegetation restoration programs, including the Natural Forest Conservation Program (NFCP), the Grain for Green Project (GGP),and the Yangtze River Basin (YZRB) shelterbelt construction project [13,14]. Therefore, under the realistic conditions of global warming and intensified human activities, it is necessary to clarify the mechanism behind vegetation change. Meanwhile, due to the complex mechanisms of vegetation change, quantifying the contributions of main drivers to vegetation changes remains a challenge.

Many studies focus on the impacts of climatic factors (temperature and precipitation) on vegetation changes, in which regression or correlation analysis has been commonly conducted [15]. In some areas, such as Sahel, Red River Basin and Inner Mongolia, precipitation plays a key role in vegetation changes [11,16], while in other areas like the Tibetan Plateau [17], vegetation changes are mainly determined by temperature. Many previous studies only focused on climatic factors, while others in recent years took climatic factors and anthropogenic factors as independent variables. Some statistical and machine learning methods have been used to identify the effects of natural and anthropogenic drivers separately [18,19]. These studies deepened our understanding of the potential causes of vegetation change and provide a reference for studying this issue. In other studies, the regression relationships between NDVI and climatic factors were established, and then NDVI residuals (defined as differences between the predicted and observed NDVI values) are considered as a consequence affected by human activities [20,21]. Furthermore, the residual analysis method is used to distinguish between the impact of climate change and ecological restoration projects on vegetation. The residual analysis is scientific and concise, and has been used widely [14,22]. Therefore, based on the residual analysis method, incorporating multiple time scale analyses into the assessment of the relative importance of climate variations and anthropogenic activities will be helpful to understand the dominant factors affecting vegetation change in different regions [19].

In recent decades, the Yangtze River Economic Belt (YREB) has experienced economic take-off and population growth. At the same time, the local climate has also changed significantly due to the impact of global warming. Specifically, the local precipitation and temperature have increased in recent decades [23,24]. The region is affected by both strong human disturbance and climate change, and the local NDVI has increased significantly. Therefore, the YREB has complete research elements, which is an excellent research area. Previous studies of the YREB were mainly focused on ecosystem services, carbon emissions and policies of ecological protection [23,25–27]. While the changes of NDVI and the driving factors behind them remain unclear, this paper aims to address this issue. It is also expected to be an exemplary for relevant research in other regions.

Therefore, this study attempts to reveal the spatial and temporal characteristics of vegetation change using the NDVI data that span the time series over the last four decades in the YREB as an example. Section 2 introduces the study area, data sources and methodology. The results are presented in Section 3. In Section 4, the potential uncertainties are discussed. Finally, the study is summarized in Section 5.

## 2. Materials and Methods

### 2.1. Study Area

The study area is the Yangtze River Economic Belt (YREB), which consists of Sichuan (SC), Yunnan (YN), Chongqing (CQ), Guizhou (GZ), Hubei (HB), Hunan (HN), Jiangxi (JX), Anhui (AH) and Zhejiang (ZJ) Provinces as well as Shanghai municipality (Figure 1). The region spans about 2400 km from east to west and about 1500 km from south to north. The YREB spans the three major regions of Eastern, Central and Western China, with a total land area of about 2.05 million $km^2$, accounting for 21.4% of the country. Its population and

GDP make up 40% of the national total [28]. The population urbanization rate of the YREB increased from 14.08% in 1978 to 59.58% in 2018, and its urban built-up areas also accounted for 40.05% of the total area of urban built-up areas in the country. The total agricultural output value of YREB accounted for 42.92% of the national total agricultural output value, while its absolute output value rose from 16,324 billion in 2010 to 27,822 billion in 2018 [29]. It is an important grain production area and the most important water conservation area as well as ecological barrier in China. Most areas are in the humid and semi-humid climate zone. The complex landscape and diverse climates make the region rich in vegetation types, which include alpine vegetation, forest and shrub land, grassland and meadow, cultivated vegetation and swamp, etc. Among them, the cultivated vegetation covers the largest area, accounting for 37.6% of the study area, and is mainly distributed in the Sichuan Basin, the Huang Huai Plain and the Jianghan Plain. The second is shrub and forest land, which account for 45.1% of the study area (Figure 1).

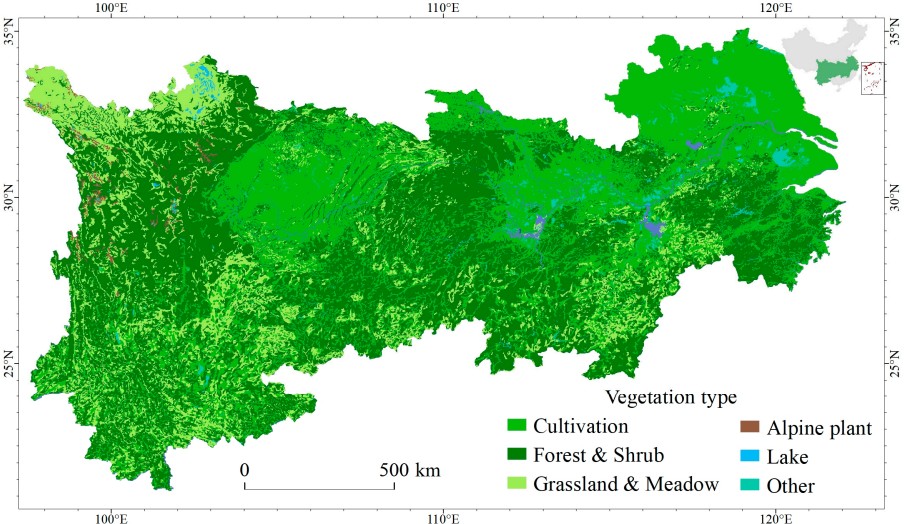

**Figure 1.** The vegetation types in the YREB (Plant science data center, Chinese Academy of Sciences (edited by Zhang Xinshi), 2007) and the location of the study area in China (green shading in the top-right corner).

### 2.2. Data Sources

### 2.2.1. NDVI Dataset

Two global NDVI datasets were used in this study. One of the NDVI datasets is derived from the Advanced Very High Resolution Radiometer (AVHRR) Global Inventory Modeling and Mapping Studies (GIMMS). The latest version of the GIMMS NDVI dataset is named NDVI3g (third generation GIMMS NDVI from AVHRR sensors). The GIMMS NDVI data were accessed from: https://ecocast.arc.nasa.gov/data/pub/gimms, accessed on 10 May 2021. It is named the GIMMS3g NDVI dataset hereinafter. The other NDVI dataset is retrieved from Moderate Resolution Imaging Spectroradiometer (MODIS) measurement data. The MODIS NDVI data were accessed from: https://ladsweb.modaps.eosdis.nasa.gov/, accessed on 13 May 2021. It is named the MODIS NDVI dataset hereinafter. The reading and preprocessing of remote sensing images are carried out in Python and processed with *gdal* package. The GIMMS3g NDVI dataset has a spatial resolution of 0.0833°, and a time interval of half a month. It spans the period from 1981 to 2015. The maximum value composite (MVC) method was used to acquire monthly NDVI data and reduce the influence of clouds and aerosols in the atmosphere [30]. This dataset has been widely used to study the long-term changes of vegetation cover in many regions [31,32].

The Terra MODIS NDVI dataset is provided by the land Distributed Active Archive Center (LP daac/nasa) for Land Processes of NASA MODIS. In this study, the MOD13a2 V6 product was used. For the purpose of minimizing the effects of noise, we retrieved the monthly NDVI with the maximal value composites (MVCs). It has a spatial resolution of

1 by 1 km, and a time interval of 16 days. This dataset was started in 2000 and has been updated until now. It is also being extensively used to study vegetation dynamics [33].

### 2.2.2. Meteorology Data

The meteorological data used in this study include monthly temperature and precipitation from 130 meteorological measurements stations. They span from 1982 to 2019 and are provided by the National Climate Data Center of China (http://data.cma.cn/, accessed on 21 June 2021).

### 2.2.3. Social and Economic Data

This study also used agricultural and economic inventory data as a measure of human activities. There are two indicators, namely, the Gross Domestic Product (GDP) per capita, and agricultural employees as a percentage of total population. These inventory data were derived from the statistical yearbooks of each province. The data span the period from 2000 to 2016. These yearbooks are available at: http://www.stats.gov.cn/tjsj/ndsj/, accessed on 6 June 2021.

### *2.3. Methods*

Firstly, we depicted the temporal changes in the NDVI and their spatial variations. Secondly, the NDVI changes were divided into two parts, driven by climate factors and non-climate factors, respectively. Thirdly, we clustered the NDVI variations driven by non-climate factors to observe the spatial characteristics. Finally, we analyzed the relationship between NDVI changes driven by non-climatic factors and human activity indicators.

### 2.3.1. NDVI Prediction in the Growing Season with Temperature and Precipitation

With the temperature and precipitation in the same period as independent variables and NDVI as a dependent variable, multiple regressions were carried out for NDVI prediction in the growing season (expression (1)). The model was calibrated, respectively, for each prefecture because it is convenient to analyze the mechanism of prefecture-level administrative divisions, and the error caused by the differences within the prefecture is acceptable. Thus, for each prefecture, we obtained a unique prediction model as well as the predicted NDVI time serial (i.e., $NDVI_{pre}$) with local climate data and residual NDVI time serial (i.e., $NDVI_{res}$). The $NDVI_{res}$ could be considered as contributions of non-climate factors and may be hence connected to human activities. The regression models and the relationship between the $NDVI_{obs}$, $NDVI_{pre}$, and $NDVI_{res}$ are as follows:

$$NDVI_{pre} = \alpha P + \beta T + \varepsilon \tag{1}$$

$$NDVI_{res} = NDVI_{obs} - NDVI_{pre} \tag{2}$$

where, $P$ and $T$ denote the standardized warm seasonal precipitation and temperature, respectively. $NDVI_{pre}$ and $NDVI_{obs}$ denote the predicted and observed NDVI, $NDVI_{res}$ denotes the residues which could not be predicted with precipitation and temperature, $\alpha$ and $\beta$ denote the regression coefficients, and $\varepsilon$ denotes the intercept from the calibrations. The construction of the multiple regression model is carried out in Python and processed with the *statsmodels* package.

### 2.3.2. Cluster Analysis of the $NDVI_{res}$ TIME Serials

Cluster analysis is a quantitative method to classify the samples. Its basic principle is to cluster the samples by quantitatively determining the closeness between samples based on their own attributes with a certain similarity or difference index. The K-means clustering method was used in this study. It has the advantages of being simple and easy to understand [34].

There are 136 prefectures in the whole study area, and each prefecture has an $NDVI_{res}$ serial. The K-means clustering analysis was carried out on the 136 $NDVI_{res}$ serials. Here,

we specified the cluster K. Firstly, the K *NDVI_{res}* serials were randomly selected as the initial cluster centers. Then, the Euclidean distance (expression (3)) to these cluster centers was calculated for each remaining *NDVI_{res}* serial. They were assigned to the class with the closest distance. Next, the new center point was calculated for each class. The process continued to iterate until the new center no longer changed. Eventually, the meanly *NDVI_{res}* serials were calculated for each class. Through comparing them with each other, the individual characters of each class of *NDVI_{res}* serials are highlighted.

$$d = \sqrt{\sum_{i=1982}^{2019} (x_i - y_i)^2}. \tag{3}$$

where $i = 1982, 1983, \ldots, 2019$ denotes the year, and $x_i$ and $y_i$ donate value of serials $x$ and $y$ in year $i$.

## 3. Results

### 3.1. Temporal Variations in NDVI and the Spatial Pattern

3.1.1. Temporal Variations of the Regional Mean NDVI Values

In the YREB, the correlation between GIMMS and MODIS data is low (Figure 2), and does not meet the requirements for fusion. According to the Mann–Kendall test on the GIMMS NDVI3g, the turning point was in 2003 (Figure 3). For the above two reasons, they are divided into two time series, one using GIMMS data from 1982 to 2003, and the other using MODIS data from 2003 to 2019. While the NDVI values of the two sets of data have a slight gap, the trend of NDVI in the common period of the two sets of data is consistent. Figure 2 shows that the regional mean growing season (May–September) NDVI in the YREB increased significantly during the period from 1982 to 2019. The rising trend of NDVI shows that the surface was greening and the vegetation coverage was improved [35]. The upward trend did not maintain a constant speed, but it occurred mainly in the 1980s and the 21st century, during which a downward trend occurred. Specifically, from 1982 to 1997, there was a peak in 1997. Then, NDVI has continued to increase since 2003 and peaked again in 2019. Between the two uptrend sessions, there was a slight declining trend. To be specific, from 1982 to 1992, when the first peak occurred, the increase rate was as high as 0.019 per decade. Then, since 2000, both the GIMMS3g and MODIS data consistently showed a prominent uptrend again. According to MODIS data, NDVI increased by 0.02 per decade from 2003 to 2019.

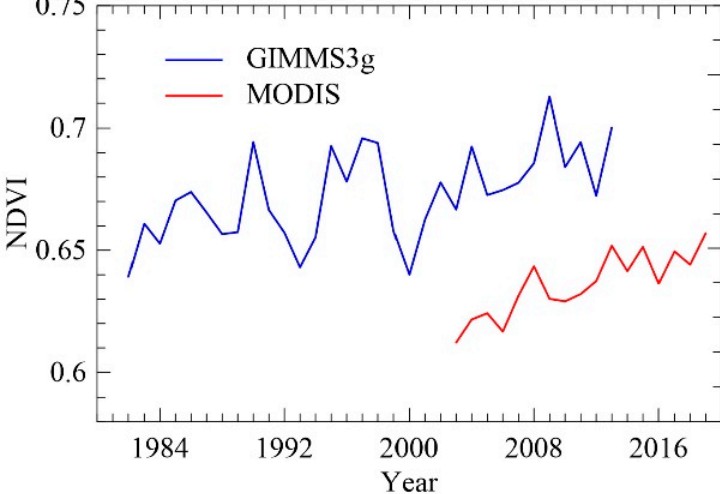

**Figure 2.** Temporal variations of growing season (May–September) NDVI during 1982 to 2019 in the YREB.

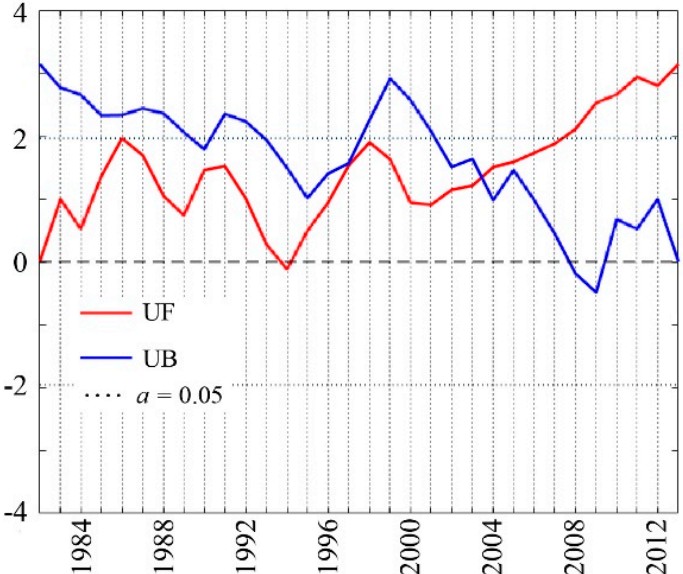

**Figure 3.** Mann–Kendall test of GIMMS3g NDVI during 1982 to 2013.

3.1.2. Spatial Variations of NDVI Trends during the Warm Season

Figure 4 shows the spatial patterns of the NDVI trends for each period. We found that the hotspots of NDVI changes shifted from one period to another. During 1982–2003, the NDVI uptrend existed over approximately 81.7% of the study area. It mainly existed in the north of Jiangsu and Anhui, Hubei, Hunan and the west of Yunnan. In the north of Jiangsu, the NDVI uptrend was the largest, with an increasing rate of up to 0.1 per decade, making it a hotspot of surface greening. This could be explained by the increase of crop yield due to the development of modern agriculture [7]. Meanwhile, the areas where NDVI decreased during this period mainly existed in the Yangtze River Delta urban agglomeration, the Wuhan urban agglomeration and the Chengdu-Chongqing urban agglomeration, among which the Yangtze River Delta urban agglomeration changed most significantly. The NDVI change rate of the Yangtze River Delta is approximately −0.07 per decade, which is the much larger than other areas. This result was consistent with the findings of Jiang et al. (2022) [35].

From 2003 to 2019, the NDVI uptrend accounted for 79.3% of the study area. The most distinct uptrend occurred at the junction of Chongqing, Guizhou and Yunnan (Figure 4b), with an increase rate of more than 0.1 per decade. Taken together, there was a distinct greening belt from Chongqing to east Yunnan province. In addition, NDVI also increased significantly in Hubei and Hunan. Meanwhile, among the extensive uptrend of NDVI, there were also NDVI decline with a rate of approximately −0.05 per decade in the western Sichuan and Yunnan as well as a larger NDVI decline with the rate of −0.1 per decade in the northern Zhejiang.

The abovementioned findings highlight the prominent greening trend over the study area from 1982 to 2019, with two distinct greening periods. During 1982–2003, NDVI generally increased throughout the whole study area except for the Chengdu Chongqing urban agglomeration, the Wuhan urban agglomeration, the Yangtze River Delta urban agglomeration and their nearby areas. This seems to indicate that the existence of an urban agglomeration will hinder the increase of NDVI [19]. During 2003–2019, the greening hotspot existed at the junction of Chongqing, Guizhou and Yunnan. The reason for this may be related to the Ecological Restoration Projects (ERPs) implemented by the Chinese government in the mountainous areas of Southwest China since the late 1990s [36]. Overall, NDVI in the whole study area generally has an uptrend, and there is no large-scale degradation except for a few metropolis areas in the Yangtze River Delta.

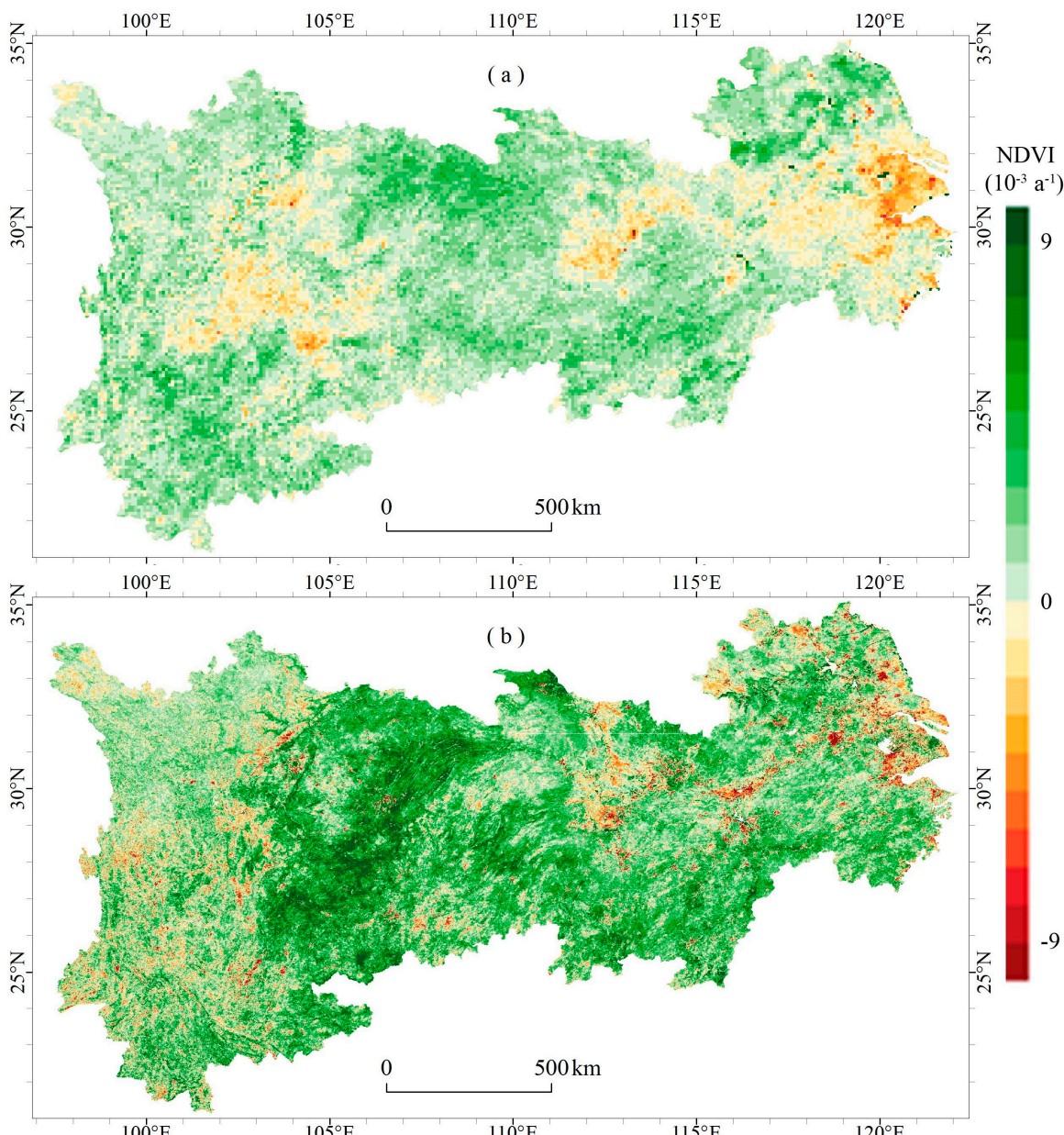

**Figure 4.** Spatial patterns of the growing season NDVI trends in YREB: (**a**) 1982–2003 and (**b**) 2003–2019.

### 3.2. Variations of NDVI with Climate Factors

Figure 5 shows the confidence levels of the NDVI prediction models with both precipitation and temperature as the predictors for each prefecture during the period from 1982 to 2003. The results from the original interannual variations (Figure 5a) show that climate factors have few effects on the changes of NDVI. In order to explore the long-term trend changes, the five-point smoothing method is used in the driving analysis below. The policies and activities of the Chinese government are mainly based on periods of five years, namely five-year plans. After removing the interannual variations by using the five-point smoothing method, the significant models exist in 64.7% of prefectures (Figure 5b), which mainly existed in the Jiangxi, Hunan and Hubei provinces. Among them, the positive correlations with precipitation account for as much as 29.4% of the prefectures and the positive correlations with temperature account for 0.177 of the prefectures. It is noted that as an exception there are also negative correlations with precipitation and temperature in, for example, the Yangtze River Delta.

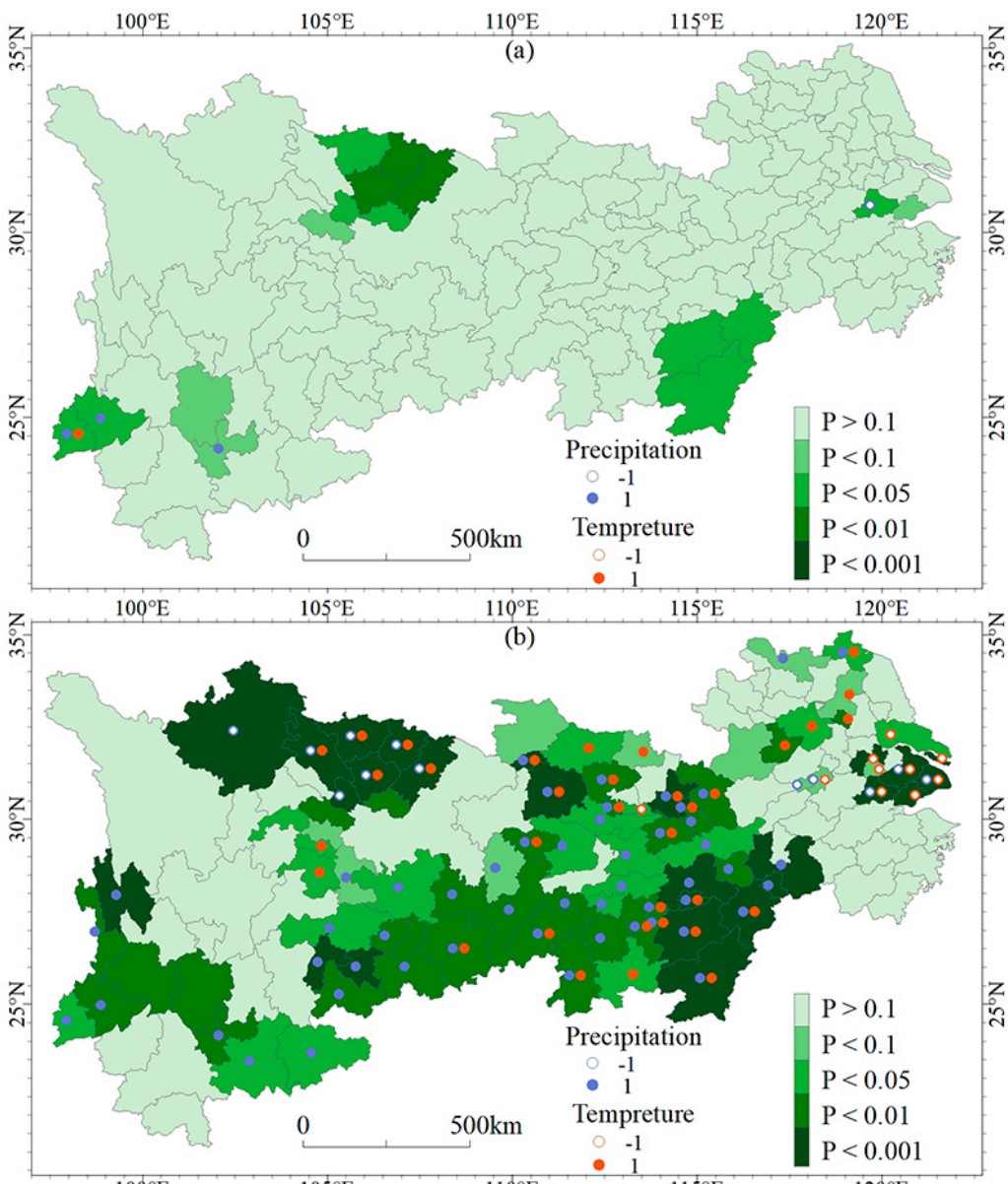

**Figure 5.** Confidence levels of the growing season NDVI prediction model with temperature and precipitation as predictors calibrated with the original annual data (**a**) and with the five-year smoothed data (**b**) from 1982 to 2003 (the circles and dots denote negative and positive correlations, respectively, with $p < 0.1$; red denotes the correlation between NDVI and temperature, and blue denotes precipitation).

Figure 6a shows that climate factors also have few effects on the interannual variations of NDVI from 2003 to 2019. After removing the interannual variations by using the five-point smoothing method, the significant models exist in 77.9% of prefectures (Figure 6b), which mainly existed in the Jiangxi, Hunan and Hubei provinces. Among them the positive correlations with precipitation account for as much as 41.1% of the prefectures and the positive correlations with temperature account for 59.5% of the prefectures. Taking together the findings presented in Figures 5 and 6, the climate factors may have little effect on the inter-annual variations of NDVI. At the time scale of longer than five years, the NDVI variations are significantly correlated with precipitation and temperature in some prefectures. From 1982 to 2003, the significant correlations with temperature took a smaller area than those with precipitation, while, from 2003 to 2019, it is the reverse [37].

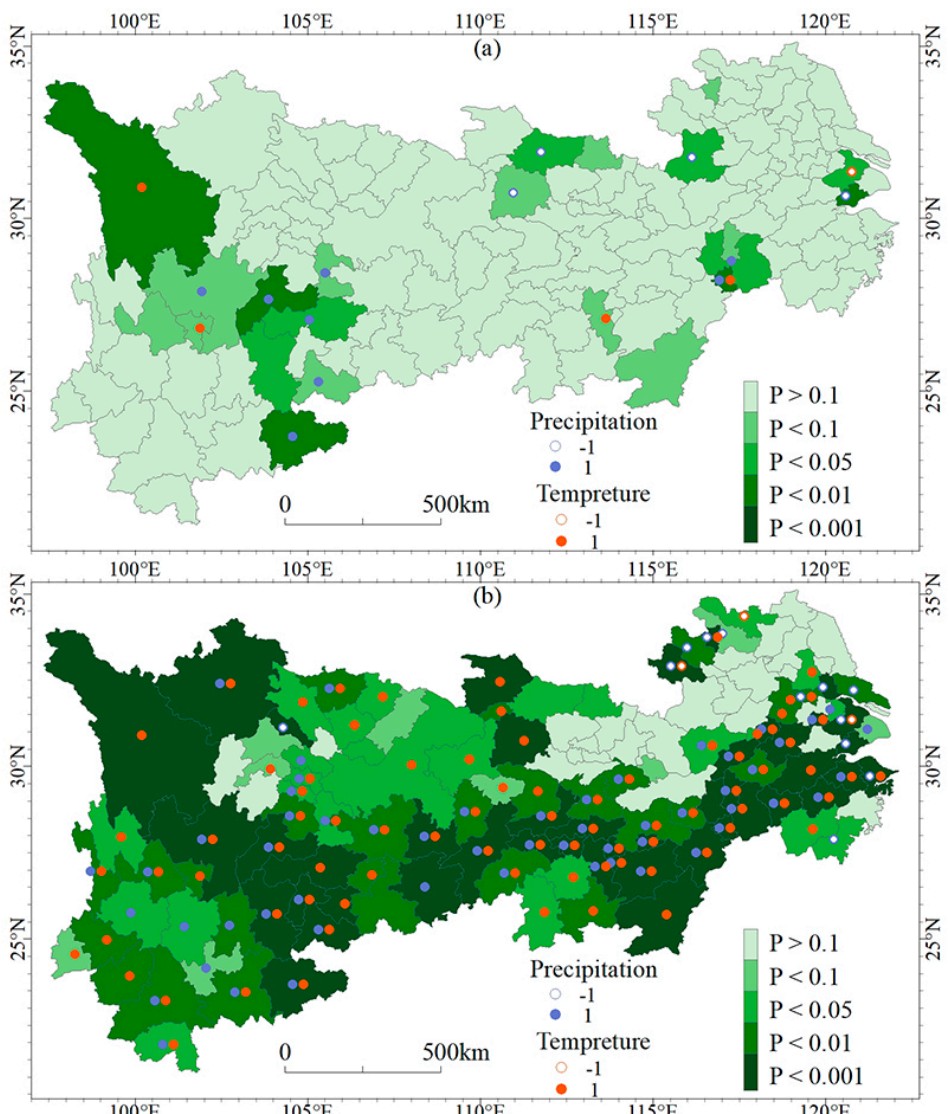

**Figure 6.** Confidence levels of the growing season NDVI prediction model with temperature and precipitation as predictors calibrated with the original annual data (**a**) and with the five-year smoothed data (**b**) from 2003 to 2019 (the circles and dots denote negative and positive correlations, respectively, with *p* < 0.1; red denotes the correlation between NDVI and temperature, and blue denotes precipitation).

Figure 7 shows the spatial patterns of $NDVI_{pre}$ slopes using the prefecture-level precipitation and temperature data as well as the $NDVI_{res}$ by subtracting the predictions from the observations. We found that the $NDVI_{pre}$ exhibited an uptrend in some prefectures; however, the slope of the uptrend was much weaker than that of the $NDVI_{obs}$, and its spatial distributions (Figure 7a) were different from the observations shown by Figure 4a. During the period from 1982 to 2003, the correlation coefficient is only 0.39 between the spatial variability of the $NDVI_{pre}$ slope and the $NDVI_{obs}$ slope, while the correlation coefficient reaches as high as 0.77 between the spatial variability of $NDVI_{obs}$ and $NDVI_{res}$ (Figure 8a,b). From 2003 to 2019, the correlation coefficient between the spatial variability of $NDVI_{obs}$ and $NDVI_{pre}$ was only 0.27, which is also much smaller than the correlation coefficient of 0.8 between the spatial variability of $NDVI_{obs}$ and $NDVI_{res}$ (Figure 8c,d). These findings suggest that the spatial variability of $NDVI_{obs}$ was more explained by that of non-climate factor-induced NDVI slopes that by the climate factors-induced NDVI slopes. Taking together the above findings, the climate changes may contribute to the NDVI uptrend in some prefectures but this could not explain such large spatial variability in the NDVI slope

over the study area. The spatial variability of the NDVI slope over the study area may be caused by non-climate factor changes. This is consistent with previous research [14].

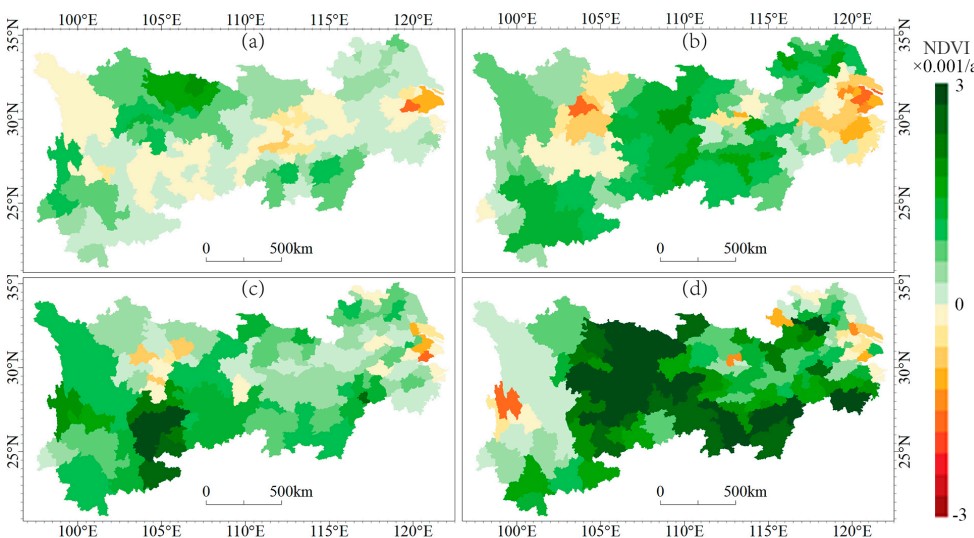

**Figure 7.** Predicted NDVI (i.e., $NDIV_{pre}$) changes using precipitation and temperature (left panel (**a**,**c**)) and its residual (i.e., $NDIV_{res}$) changes (right panel (**b**,**d**)) for periods from 1982 to 2003 (top panel (**a**,**b**)) and from 2003 to 2019 (bottom panel (**c**,**d**)).

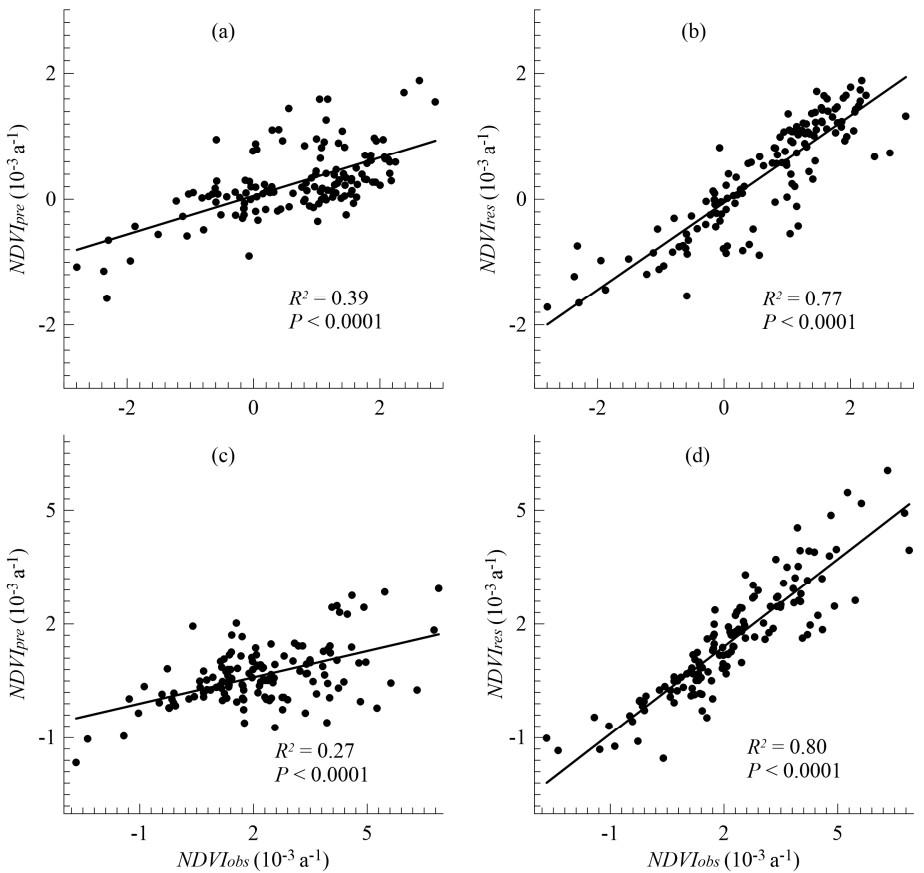

**Figure 8.** Scatter plots of the observed NDVI changes against the predicted NDVI (i.e., $NDVI_{pre}$) changes using the precipitation and temperature (left panel (**a**,**c**)) and its residuals ($NDVI_{res}$, right panel (**b**,**d**)) at the prefecture level from 1982 to 2003 (top panel (**a**,**b**)) and from 2003 to 2019 (bottom panel (**c**,**d**)).

### 3.3. The Modes of the Changes in the $NDVI_{res}$

The main modes of $NDVI_{res}$ derived from cluster analysis and their spatial patterns for the 1982–2003 and 2003–2019 periods are, respectively, shown by Figures 9 and 10. From 1982 to 2003, the $NDVI_{res}$ changes could be grouped into four classes. Class I is mainly characterized by a decline trend. It mainly existed in the Yangtze River Delta urban agglomeration and in the Sichuan Basin Chengdu-Chongqing urban agglomeration, which underwent rapid urbanization with conversion from cropland to building area. Class II is mainly characterized by approximate trendlessness. It mainly existed in the mountainous areas of Sichuan and Yunnan, with a low population density and low urbanization. Class III is mainly characterized by uptrend in 1980s and a slight uptrend during 1990s. It mainly existed in Hubei and Hunan Province as well as Chongqing Municipality. Class IV is mainly characterized by a prominent uptrend before 1997 and a decline thereafter. It mainly existed in northern partitions of Jiangsu and Anhui provinces, which is a typical agriculture area of China.

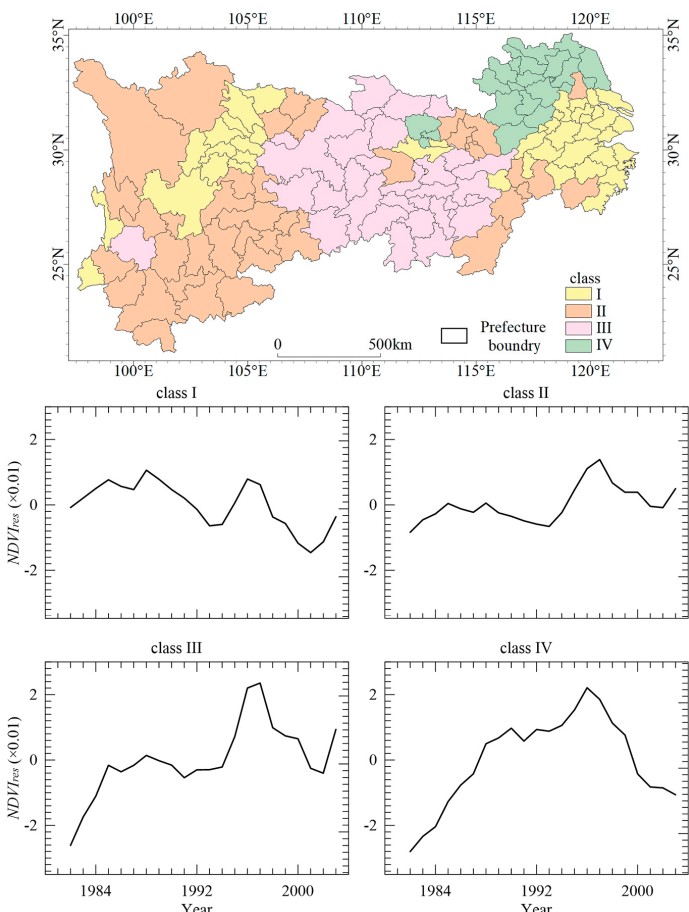

**Figure 9.** Spatial pattern of $NDVI_{res}$ serials classification (above) and temporal variations of $NDVI_{res}$ of each class (below) from 1982 to 2003.

From 2003 to 2019, the $NDVI_{res}$ changes could be grouped into six classes. Classes I–III share a common character, which is uptrend. For Class I, the uptrend likely started around 2010. The uptrend slope of Class I is smaller than Classes II and III. For both Classes II and III, the uptrend likely started around 2005. The uptrend slope of Class II is much larger than that of Class III. Class I mainly exists in the northern partition of the Sichuan Basin and the main body of Hunan and Jiangxi provinces. Class II mainly exists in the Chongqing Municipality and Class III mainly exists in the Guizhou Province. The above areas of Classes I–III are also the hotspots where the forest protection policy was implemented [38]. Class IV is mainly characterized by trendlessness. It mainly existed

in the western partition of Sichuan Province and main body of Yunnan Province as well as area surrounding the Yangtze River Delta. Class V is mainly characterized by a weak uptrend and mainly exists in main body of Hubei Province as well as Jiangsu and Anhui provinces, which are dominantly covered by arable land. Class VI is mainly characterized by trendlessness prior to 2010 and a fast decline thereafter. It takes a small area only in the norther partition of Anhui Province.

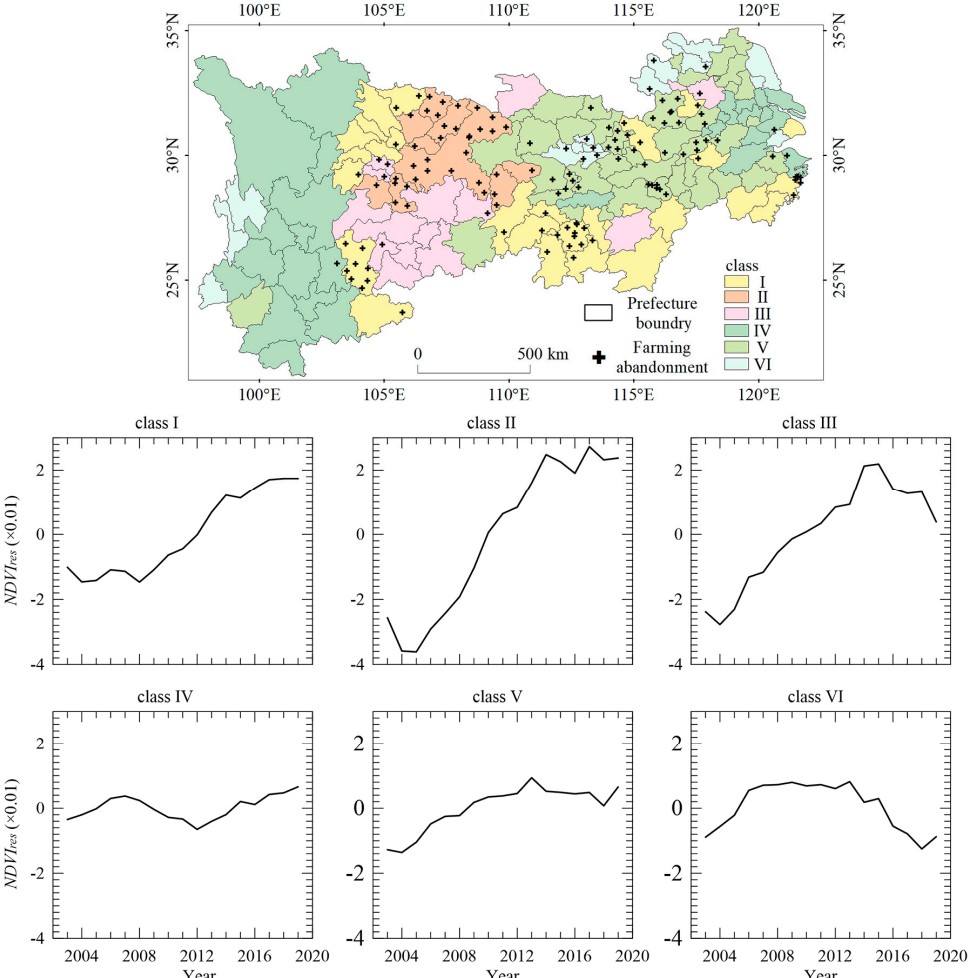

**Figure 10.** The same as Figure 9 but for the $NDVI_{res}$ serials from 1982 to 2003, as well as with black crosses in the top panel indicating cropland abandonment from 1992 to 2017 [39].

### 3.4. The Agriculture Status behind the $NDVI_{res}$ Uptrend from 2003 to 2019

Agriculture practices, as an important human activity, broadly affect vegetation coverage. In the last decade, cropland abandonment arose in the study area. Zhang et al. (2019) compiled the cropland abandonment records from 112 published papers based on field surveys [39]. Figure 10 shows the majority of abandonment records (85.1%) exist in Classes I, II and V of the areas in Sichuan, Chongqing, Hubei, Hunan and Anhui provinces. Among them, there are 27.3% in Class I, 27.3% in Class II and 30.5% in Class V. As illustrated by Figure 10, the $NDVI_{res}$ in Classes I, II and V are commonly characterized by an uptrend. These findings suggest that cropland abandonment may co-exist with an NDVI uptrend.

There are many factors for land abandonment, and the main reasons are low agricultural income and shortage of labor forces [39]. Local economic development and industrial transformation are the economic basis of cropland abandonment. For example, the per capita GDP of the region increased rapidly between 2002 and 2011. The average annual growth rates in major abandoned areas of Zhejiang, Anhui, Jiangxi, Hubei, Hunan, Sichuan and Chongqing were 14.1%, 14.2%, 15.4%, 15.5%, 14.9% and 15.1%, respectively (Table 1).

It can be seen that the local socio-economic situation is in a period of rapid economic development. From the perspective of agricultural employees, there is a significant decline in all provinces. For example, the proportion of agricultural employees in Anhui and Chongqing has decreased by 43.3% and 43.9%, respectively (Table 2). At the same time, these provinces also have frequent abandonment of farming. To sum up, economic development may have led to local economic transformation, while the shift of agricultural labor forces has led to abandonment of farming.

**Table 1.** Gross Domestic Product (GDP) per capita ($10^3$ Yuan).

| Province<br>Year | SC | CQ | GZ | HB | HN | JX | AH | ZJ |
|---|---|---|---|---|---|---|---|---|
| 2002–2006 | 8.02 | 10.84 | 4.59 | 10.13 | 9.24 | 8.23 | 7.68 | 23.82 |
| 2007–2011 | 18.62 | 24.43 | 11.66 | 24.2 | 21.61 | 18.79 | 17.89 | 46.61 |
| 2012–2016 | 33.53 | 48.04 | 26.55 | 46.85 | 38.42 | 33.04 | 34.06 | 73.58 |

**Table 2.** Proportion of agricultural employees in the total population (%).

| Province<br>Year | SC | CQ | GZ | HB | HN | JX | AH | ZJ |
|---|---|---|---|---|---|---|---|---|
| 2000 | 77.6 | 72.8 | 81.8 | 66.4 | 75 | 68.4 | 75 | 33.7 |
| 2010 | 60.2 | 50.3 | 68.9 | 53.1 | 55.9 | 44.4 | 54.2 | 14.7 |
| 2016 | 38.6 | 28.9 | 57.3 | 36.8 | 40.7 | 30 | 31.7 | 12.4 |

Generally, large-scale abandonment of farming will lead to the reduction of surface vegetation, but the NDVI in these areas shows an increasing trend. The reason may be that a large number of labor forces are transferred out, the disturbance of local vegetation by farmers is suddenly weakened, local vegetation grows naturally under locally appropriate moisture and heat conditions and vegetation grows better. On the other hand, most of the abandoned cultivated land is sloping cultivated land with high input and low output. For example, Sichuan Province has a total of 4520 km$^2$ of cropland, 44.5% of which is located in the depression areas with a slope < 5° and 17% on slopes > 25°. A large amount of sloping farmland was returned to forest, and the surface vegetation increased significantly [40,41]. Meanwhile, the Chinese government began to implement the Natural Forest Protection Program (NFPP) and the Sloping Land Conversion Program (SLCP) at the beginning of the 21st century [7]. Therefore, under the joint impact of economic development and environmental policies, the area with high vegetation cover increased significantly [42], and the NDVI also increased significantly.

## 4. Discussion

As mentioned above, there has been a significant greening trend across the YREB over the past four decades. In particular, the surface greening during 2003–2019 was much more prominent than that during 1982–2003. Meanwhile, the surface greening mainly occurred in the mountain areas in Sichuan-Chongqing-Guizhou as well as Hunan provinces. Keeping with existing studies [10], temperature and precipitation changes may partly contribute to surface greening [11]. However, the spatial variabilities of surface greening were little explained by climate-induced NDVI changes, while they are mostly explained by NDVI changes caused by non-climate factors. These prominent surface greenings co-existed with cropland abandonment, which is derived from industrial development and, hence, labor transition from primary industrial to the second and third industries [39].

It is undeniable that this study has some deficiencies. First, due to the defects of NDVI, it may be worth trying to use other vegetation indices as aids to improve the accuracy, such as EVI and SAVI. In addition, NDVI will also be affected by soil background. Specifically, the NDVI value of dry soil is lower than that of wet soil [43]. When there are great differences in soil moisture in the study area, large errors may occur. However, in the YREB, the soil background has little effect on NDVI and will not affect the experimental

results. Because the YREB belongs to China's humid and semi humid areas, it is generally rainy in the growing season and the river network is dense. It can be considered that the YREB is a relatively moist soil background. The impact of soil can be ignored in this study.

Due to the small amount of data, this paper does not have quality control preprocessing to filter out some low-quality remote sensing data. However, Maximum Value Composite (MVC) was carried out to eliminate disturbances created by cloud contamination, atmospheric variability, bi-directional effects and sensor malfunctions. Here we focus on those stationary changes and trends between years over the long time series and the processed NDVI data meets our requirements. Our research results are consistent with previous scholars [14,19], which is reasonable. Therefore, based on the above reasons, the impact of data quality in the result can be ignored.

In addition, this study only diagnoses and compares the sensitivity between anthropogenic factors and climate to NDVI changes. The complexity of the interaction between human and vegetation cannot be ignored. For example, Lin et al. (2021) adopted SEM modeling to better understand and explain the complex interrelationships behind the spatial temporal change of NDVI [7]. However, the accuracy of the model needs to be improved. More study on non-linear relationships between the driving factors and NDVI change should be conducted. In addition, in order to understand the interaction between humans and vegetation, it is worth collecting more climatic data and socio-economic data, so that more factors can be considered in the study. Finally, taking a prefecture as a pixel may ignore the internal differences of physical geographic and socio-economic factors. This may make our results inaccurate. In order to overcome this deficiency, it is valuable to collect county-level socio-economic data and use a county as a pixel in the future.

The reasons for the difference between GIMMS NDVI and MODIS NDVI are complex, some of which are as follows: There are several aspects of AVHRR sensor design that are not ideal for vegetation trend studies, such as post-launch degradation in sensor calibrations and drift in the satellite overpass times. The seasonal variations in sun-sensor viewing geometry (as defined by the Bidirectional Reflectance Distribution Function, BRDF) combined with sensor drift over time have a large effect on the time series of observed NDVI for a given location. Additionally, the spectral configuration of the AVHRR sensors (number of bands, wavelengths covered and the specific band Spectral Response Functions (SRFs)) does not permit an accurate atmospheric correction scheme to be applied and absorption and scattering by atmospheric components such as water vapour are a source of error in AVHRR estimates of surface NDVI. The AVHRR channel 2 (near-infrared band) covers wavelengths in which there is considerable absorption by water vapour in the atmosphere, influencing observed NDVI. Therefore, the noise of GIMMS NDVI data cannot be neglected. Moreover, the near-infrared band and the red band difference between GIMMS and MODIS will inevitably lead to the difference of NDVI. It is a complex project to correct the two sets of data, and it is beyond the range of our ability.

It is worth noting that there are more climatic factors than temperature and precipitation that can significantly regulate the vegetation growth. For instance, it is widely reported that $CO_2$ in the air plays an important role in vegetation growth [44,45]. There is no doubt about this conclusion. The continuous increase in airborne $CO_2$ concentration over the last four decades cannot be ignored. Increased atmospheric $CO_2$ concentration can affect vegetation in many ways. For example, the increased carbon supply can allow for higher photosynthetic rates and therefore more plant growth [46]. Moreover, the increased carbon supply can allow plants to maintain photosynthetic rates with lower conductance, thereby lowering the amount of water lost during gas exchange with the atmosphere [46,47]. In addition, solar radiation also has an important impact on vegetation growth [48], and topography and soil will indirectly affect vegetation conditions by affecting human activities [7]. This reminds us that we can collect more natural factors to improve simulation accuracy in the future.

## 5. Conclusions

The above results show that the YREB has had a significant greening trend in the past four decades. The greening trend is divided into two periods: 1982–2003 and 2003–2019. The greening trend from 1982 to 2003 generally increased except for the three major urban agglomerations, and rapid urbanization may be the main reason for the decrease of NDVI. Most areas showed an obvious greening trend from 2003 to 2019. Analysis shows that anthropogenic factors play a dominant role in NDVI changes; the vegetation conditions in the western hilly areas have been greatly improved, driven by changes in industrial structure and policies. Meanwhile, climate factors cannot be ignored, and temperature is particularly important in the process of NDVI increase from 2003 to 2019.

This study supports the important role of human activities in vegetation changes in the YREB. Moreover, from studies with longer time scales and larger spatial domains, the main anthropogenic factors affecting NDVI may vary from region to region. This means that we should pay attention to the anthropogenic domain, adjust measures to local conditions and pay attention to the division of regions and classification studies in the future. More abundant human factors should be used to improve the accuracy of the model. Parameterizing human factors under the currently popular surface model will be a potential improvement to the land vegetation dynamic model [49]. At the same time, it also has reference value for studying the impact of greening hotspot transfer on regional surface energy and atmospheric water flux. This is of great reference value for understanding the causes of regional climate change, because land vegetation dynamics can feed back to local and regional climate by regulating surface energy and water flux to the atmosphere [50].

While most studies mainly rely on rough spatial resolution data such as MODIS or NOAA-AVHRR, in the future, the research on the complex and spatial change process behind vegetation changes will benefit from the availability of high-resolution satellites such as Sentinel-2, which has been operational since June 2015. This satellite provides new prospects for long- and short-term monitoring in the world. In particular, by providing frequent and high-quality observations in time series, we hope that this information will become more readily available in the near future, so as to benefit from a wide range of vegetation problems.

**Author Contributions:** Conceptualization, X.Z. and H.L.; methodology, X.Z. and L.Z.; software, L.Z.; validation, L.Z. and X.Z.; formal analysis, L.Z.; investigation, H.L.; resources, X.Z.; data curation, L.Z.; writing—original draft preparation, L.Z.; writing—review and editing, X.Z. and X.Z.; visualization, L.Z.; supervision, X.Z.; project administration, X.Z.; funding acquisition, H.L. All authors have read and agreed to the published version of the manuscript.

**Funding:** This research was funded by the National Key Research and Development Program of China, grant number 2019YFA0606600; and the National Natural Science Foundation of China, grant number 41790424.

**Institutional Review Board Statement:** Not applicable.

**Informed Consent Statement:** Not applicable.

**Data Availability Statement:** The GIMMS NDVI3g dataset is available through http://poles.tpdc.ac.cn/en/data/9775f2b4-7370-4e5e-a537-3482c9a83d88/ (accessed on 10 May 2021). MODIS NDVI is downloaded from: https://lpdaac.usgs.gov/products/mod13a1v006/ (accessed on 13 May 2021). The meteorological data are provided by the National Climate Data Center of China (http://data.cma.cn/, accessed on 21 June 2021). the statistical yearbooks are available through website: http://www.stats.gov.cn/tjsj/ndsj/, accessed on 6 June 2021.

**Conflicts of Interest:** The authors declare no conflict of interest.

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
