# Peer review of "Land-Greening Hotspot Changes in the Yangtze River Economic Belt during the Last Four Decades and Their Connections to Human Activities"

_land, doi:10.3390/land11050605_

Round 1

Reviewer 1 Report

The authors present an interesting study examining vegetation greenness and drivers, including human influences. The concept and approach are sensible. However, there are some gaps in the data processing and methods that must be addressed clearly in the manuscript to better judge the results of the study. If the authors address these issues, I look forward to reading again andgetting the paper on its way to publication.

Line 60 – “affected by” or “affecting humans”

Lines 68-71: These sentences should be improved. The idea to be conveyed in these two sentences is that, despite important changes in the region, little research has been carried out or maybe no research. If the authors claim no research, please be certain. If the area is understudied, maybe mention some examples of what has been studies.

Section 1.2 Study area: Several statements, such as the economic growth statistic and agricultural production, should have citations. It would be beneficial if there were a few details added regarding the study area to give the reader a better sense of the context. Perhaps some very brief details could explain the timeline for some of the changes?

Section 2.2.1-2.2.3

This section needs revision to include the processing details. The information about the GIMMs3g data can be fairly easily looked up by the reader, but there are not enough details about the MODIS data obtained by the authors. The authors must give far more details about how they obtained, preprocessed, and generated mosaics, for example. Did they apply quality flags? How are they applied? Similarly, details about the other datasets relevant to the quality, frequency, processing of these data need to have further details. A few pieces of information are included in the methods, but not enough to construct a clear understanding of the most important steps to assure good data handling.

Section 2.3 Methods

The authors present the models used and a satisfactory explanation for the structure of these models. Depending on the detail added to the previous section, there is a need for some additional information. What programs or languages were used for data handling, processing? Please consult a few highly cited papers (some of the references used in this paper) to see how the methods include more details. Much of that information should be included here.

Results

Many of the results are interesting and seem reasonable. The writing in this section is more detailed than previous sections. The figures are compelling, although Figure 9 needs a more detailed caption, perhaps with labels for each panel and appropriate descriptions. However, it is difficult to analyze the results (or discussion and conclusion) properly without knowing more about the datasets and methods. For instance, if QA/QC was not performed on MODIS data, then NDVI are not reliable. In fact, the discrimination of hotspots of NDVI is an artifact of MODIS data that are not assured of quality for pixels. If one assumes this has been completed beforehand, the results are much more compelling.

Discussion

Again, assuming the details are correctly included in the methods, much of the discussion os compelling and interesting. In lines 364-370, the discussion of CO2 and greening would need some clarification. It is not in debate about rising atmospheric CO2 concentration, however there is some debate about the effects on greening and a more complex debate about the rate and timing of greening attributed to changing CO2 concentrations. In short, the paragraph reads as if all factors are resolved, understood and agreed upon. If a more thorough description of this were included here, it actually increase the importance of this paper and studies such as these, which add evidence to the body of literature.

Conclusions

Some modifications to the conclusions should include a “softening” of the approach to the shortcomings of the research. Potentially, the language describing these should be more about items that could not be addressed or will be addressed in future research. Perhaps this might go into the discussion rather than in the last paragraph seen by the reader. If the authors are going to suggest testing with other indices, then EVI should be among the candidates.

Author Response

Point 1: Line 60 – “affected by” or “affecting humans”

Response 1: Modified at line 71.

Point 2: Lines 68-71: These sentences should be improved. The idea to be conveyed in these two sentences is that, despite important changes in the region, little research has been carried out or maybe no research. If the authors claim no research, please be certain. If the area is understudied, maybe mention some examples of what has been studies.

Response 2: Modified at line 84-86.

Point 3: Section 1.2 Study area: Several statements, such as the economic growth statistic and agricultural production, should have citations. It would be beneficial if there were a few details added regarding the study area to give the reader a better sense of the context. Perhaps some very brief details could explain the timeline for some of the changes?

Response 3: Added at line 102-109.

Point 4: Section 2.2.1-2.2.3 This section needs revision to include the processing details. The information about the GIMMs3g data can be fairly easily looked up by the reader, but there are not enough details about the MODIS data obtained by the authors. The authors must give far more details about how they obtained, preprocessed, and generated mosaics, for example. Did they apply quality flags? How are they applied? Similarly, details about the other datasets relevant to the quality, frequency, processing of these data need to have further details. A few pieces of information are included in the methods, but not enough to construct a clear understanding of the most important steps to assure good data handling.

Response 4: Supplemented at line 127-130 and line 140-141.

Point 5: Section 2.3 Methods  The authors present the models used and a satisfactory explanation for the structure of these models. Depending on the detail added to the previous section, there is a need for some additional information. What programs or languages were used for data handling, processing? Please consult a few highly cited papers (some of the references used in this paper) to see how the methods include more details. Much of that information should be included here.

Response 5: Supplemented at line 131-132 and line178-179.

Point 6: Results  Many of the results are interesting and seem reasonable. The writing in this section is more detailed than previous sections. The figures are compelling, although Figure 9 needs a more detailed caption, perhaps with labels for each panel and appropriate descriptions. However, it is difficult to analyze the results (or discussion and conclusion) properly without knowing more about the datasets and methods. For instance, if QA/QC was not performed on MODIS data, then NDVI are not reliable. In fact, the discrimination of hotspots of NDVI is an artifact of MODIS data that are not assured of quality for pixels. If one assumes this has been completed beforehand, the results are much more compelling.

Response 6: Your comments is very pertinent. The quality of NDVI data will indeed affect the analysis results. However, due to the small amount of data, this paper does not have quality control preprocessing to screen out some low-quality remote sensing data. However, what we emphasize is the trend analysis of the whole research area, and our research results are consistent with the research results of previous scholars, which is reasonable.

Point 7: Discussion  Again, assuming the details are correctly included in the methods, much of the discussion os compelling and interesting. In lines 364-370, the discussion of CO2 and greening would need some clarification. It is not in debate about rising atmospheric CO2 concentration, however there is some debate about the effects on greening and a more complex debate about the rate and timing of greening attributed to changing CO2 concentrations. In short, the paragraph reads as if all factors are resolved, understood and agreed upon. If a more thorough description of this were included here, it actually increase the importance of this paper and studies such as these, which add evidence to the body of literature.

Response 7: Relevant content has been added at line 430-437.

Point 8: Conclusions  Some modifications to the conclusions should include a “softening” of the approach to the shortcomings of the research. Potentially, the language describing these should be more about items that could not be addressed or will be addressed in future research. Perhaps this might go into the discussion rather than in the last paragraph seen by the reader. If the authors are going to suggest testing with other indices, then EVI should be among the candidates.

Response 8: Moved to discussion at line 408-410.

Thank you for your review and many valuable comments. We look forward to hearing from you regarding our submission. We would be glad to respond to any further questions and comments that you may have.

Reviewer 2 Report

The submitted research work aims to investigate the spatial patterns of the NDVI changes in the Yangtze River Economic Belt (YREB) and their potential causes during the last four decades. Authors combined GIMMS3g NDVI data and MODIS NDVI data, and implemented the Mann-Kendall test, regression analysis with cluster analysis to diagnose the potential causes of the NDVI changes. The undertaken work is very interesting and can be considered novel and will help us understand the utility of EO data in quantifying natural phenomena. I found this paper very interesting where several technical aspects were nicely implemented and explained sufficiently. Undoubtedly, authors invested huge amount of time and have made a great effort to produce this high-quality of research which is clearly structured and the language used is largely appropriate. As final decision, I see that this manuscript in its form and level deserves to be accepted for publication in MDPI-WATER BUT after addressing below MINOR COMMENTS.

COMMENTS:

  • The title of the paper and the abstract look great.
  • The content of the paper and its design match the international standards.
  • I would suggest making fig.01 bigger; the same size with fig.4.
  • Please consider to deliver high quality of figures.
  • I strongly suggest for the authors to find the way to merge results with discussion and make it one section named “results and discussion” because I found the discussion small and tells information that can be integrated within the results section easily. .
  • I recommend for the authors to mention their suggestion about the Sentinel2 data in the conclusion section rather than in the discussion section.
  • As final general comment, please make sure to define ALL the acronyms form their first appearance in the paper. Also, all the references MUST BE CHECKED and formatted as required by MDPI- RS, also make sure that all the references have DOI number unless it is not available.

Author Response

Point 1: I would suggest making fig.01 bigger; the same size with fig.4.

Response 1: Modified on line 117.

Point 2: Please consider to deliver high quality of figures.

Response 2: Modified on line 117 and line 311.

Point 3: I strongly suggest for the authors to find the way to merge results with discussion and make it one section named “results and discussion” because I found the discussion small and tells information that can be integrated within the results section easily.

Response 3: Your suggestion is very pertinent, we adjusted and supplemented the discussion on line397-441.

Point 4: I recommend for the authors to mention their suggestion about the Sentinel2 data in the conclusion section rather than in the discussion section.

Response 4: Modified on line 464-467.

Point 5: As final general comment, please make sure to define ALL the acronyms form their first appearance in the paper. Also, all the references MUST BE CHECKED and formatted as required by MDPI- RS, also make sure that all the references have DOI number unless it is not available.

Response 5: Modified on line 13, line 16, line 39, line 53. And DOI is added in corresponding position.

Thank you for your review and many valuable comments. We look forward to hearing from you regarding our submission. We would be glad to respond to any further questions and comments that you may have. The revised manuscript is in the attachment.

Reviewer 3 Report

Dear authors,

thanks for paper. I think, that topic is interesting, but I have number of comments and I expect, that additional experiments or clarifications are needed.

  1. Introduction need to be extended and there is necessary to provide deeper analysis of previous studies. This topic is studied very often and analysis need to be extended.
  2. I see as large problem combination of GIMSS and MODIS data. You are writing, that there is low correlation between this to data sources (line 174 “In the YREB, the correlation between GIMMS and MODIS data is low“), but immediately in this paragraph you wrote, that you used data from GIMMS for period 1982 - 2003 and for period 2003  - 2003 you used data from MODIS. How this measurement can be compared.
  3. There is not clear, how social and economical data are integrated into analysis
  4. Role of clustering methods is not well clarified in paper
  5. There is necessary explain better, how climatic data was compared with NDVI indexes. Probably this need to me part of methodology, which is purely described
  6. Again is not cleat, how predictions are calculated, again is needed to add this to methodology

Author Response

Point 1: Introduction need to be extended and there is necessary to provide deeper analysis of previous studies. This topic is studied very often and analysis need to be extended.

Response 1: Modified on line 44-48, line 50-54, line 72-77, line 84-86.

Point 2: I see as large problem combination of GIMSS and MODIS data. You are writing, that there is low correlation between this to data sources (line 174 “In the YREB, the correlation between GIMMS and MODIS data is low“), but immediately in this paragraph you wrote, that you used data from GIMMS for period 1982 - 2003 and for period 2003  - 2003 you used data from MODIS. How this measurement can be compared.

Response 2: GIMMS data is used in the first period and MODIS data is used in the second period for separate analysis. We only explored the change trend and did not compare the NDVI values of the two sets of data. Supplemented on line 205-206.

Point 3: There is not clear, how social and economical data are integrated into analysis.

Response 3: Your discussion is very pertinent. Establishing the relationship between socio-economic factors and NDVI is indeed one of the key points of this kind of research. In this study, there is no direct model on the relationship between human socio-economic indicators and NDVI, but it is discussed based on the previous research results, the general situation of cropland abandonment and the local economic background. Secondly, many currents models, such as structural equation model, geographic detector and generalized additive model, are not accurate in this research, and there is no effective model, the amount of data is small, so they do not build a model to establish the relationship between socio-economic indicators and NDVI.

Point 4: Role of clustering methods is not well clarified in paper.

Response 4: Modified on line 181-183. There are 136 prefectures in the Yangtze River Economic Belt. We can't show the changes of NDVIres of each prefecture. We can only classify the prefectures with the same serial changes into four or six classes by cluster analysis, so as to simplify the steps. That is, using 4 or 6 pairs of average NDVIres serial diagrams of various categories can describe the situation of the Yangtze River Economic Belt.

Point 5: There is necessary explain better, how climatic data was compared with NDVI indexes. Probably this need to me part of methodology, which is purely described.

Response 5: Supplemented on line 164-170.

Point 6: Again is not cleat, how predictions are calculated, again is needed to add this to methodology.

Response 6: Firstly, the regression model is obtained by multiple regression using meteorological data and NDVI data, as shown in expression 1 (line 172), and then the predicted NDVI is calculated according to the model through the meteorological data of each prefecture. I don't know if I have explained clearly and look forward to your reply.

Thank you for your review and many valuable comments. We look forward to hearing from you regarding our submission. We would be glad to respond to any further questions and comments that you may have. The revised manuscript is in the attachment.

Reviewer 4 Report

The article presents an article that shows changes in land-greening hotspots in the Yangtze River Economic Belt and its relationship with human activities.

The topic of the paper is suitable for the Land journal, and the article presents an interesting study. However, the article is unclear to this reviewer on some specific points.

Why do authors select the Normalized Difference Vegetation Index (NDVI) as an indicator of vegetation growth? Why did they not use other indices such as EVI, for example? The authors should justify this selection even when they used ready-to-work datasets.

The authors should link their findings with previous scientific literature.

One of the main elements that affect NDVI and spectral information captured using satellites is the soil. Probably, there are different kinds of soils in the study area, and the authors should explain how it affects NDVI. Moreover, precipitation and temperature indeed affect NDVI, but soil strongly determines the vegetation and, therefore, NDVI, so it is another critical point that should be explained. NDVI values are relatively stable throughout the year; however, they can vary significantly due to weather conditions (for example, dry soil versus wet soil).

Finally, an extensive English language review is necessary throughout the entire text. For example, in line 111: “it [has] also been”

Specific comments:

Line 36

“among them, NDVI is the most widely used vegetation index”

It is important to explain why the authors used NDVI and not other indices. The authors reference an article (Cai et al., 2022) that only uses NDVI, and they do not justify if (or why) the NDVI is the most used index. A correct reference that supports this statement is needed.

Line 64-66

“the local climate has also changed significantly due to the impact of global warming”

In what sense? References are needed.

line 68

“Therefore, the YREB has complete research elements, which is unique in the world and few scholars have notice that. It is an excellent research area where the changes of NDVI and the driving factors behind them can be studied”

This statement is not important in a scientific article. Of course, each area is “unique” globally, and almost every article could include this. If it is so unique, it should be sustained by several references.

Moreover, if this area is so “unique”, how could this study could be extrapolated to be used in other areas? Citing the authors: “It can also play an exemplary role for relevant research in other regions.”

Line 99-105

Authors should cite the datasets properly. Moreover, they should explain the instruments. i.e. AVHRR.

Line 149

“cluster analysis […] multielement things”.

“Things” is an inappropriate word in this context. Moreover, the authors should explain this point better.

Line 174

“correlation between GIMMS and MODIS data is low”

Why the data between the two datasets it is not consistent?

Line 176

And what happened in 1997?

The authors should explain the forward and backward curves.

Line 235-245

Since these findings are not new (NDVI is well-correlated with temperature and precipitation), authors should link their work with previous scientific literature. It is not noted in the discussion. For example:

Wang, P. M. Rich & K. P. Price (2003) Temporal responses of NDVI to precipitation and temperature in the central Great Plains, USA, International Journal of Remote Sensing, 24:11, 2345-2364, DOI: 10.1080/01431160210154812

This is only an example, authors may not include it, but there are many others.

Figure 5-6

Why did the authors select five-year smoothed data and not other?

Line 269

“the uptrend was much weaker” Why? It should be explained in the discussion.

Line 344

It should be interesting to show a slope map to justify these statements.

Line 366

The authors don’t show any data that supports the statements regarding CO2.

Line 372

Authors should explain the platforms in the material and methods section or introduction.

Line 403

“LAI” is not a vegetation index. It can be estimated using vegetation indices, but it’s not one of them. Authors should re-write conclusions.

Author Response

Point 1: Why do authors select the Normalized Difference Vegetation Index (NDVI) as an indicator of vegetation growth? Why did they not use other indices such as EVI, for example? The authors should justify this selection even when they used ready-to-work datasets.

Response 1: Modified on line 39-41.

Point 2: The authors should link their findings with previous scientific literature.

Response 2: Supplemented on line 68-78, line 80-82, line 84-88.

Point 3: One of the main elements that affect NDVI and spectral information captured using satellites is the soil. Probably, there are different kinds of soils in the study area, and the authors should explain how it affects NDVI. Moreover, precipitation and temperature indeed affect NDVI, but soil strongly determines the vegetation and, therefore, NDVI, so it is another critical point that should be explained. NDVI values are relatively stable throughout the year; however, they can vary significantly due to weather conditions (for example, dry soil versus wet soil).

Response 3: New discussion section added on line 410-416.

Point 4: Finally, an extensive English language review is necessary throughout the entire text. For example, in line 111: “it [has] also been”.

Response 4: I have checked the full text and revise.

Point 5: Line 36:“among them, NDVI is the most widely used vegetation index”

It is important to explain why the authors used NDVI and not other indices. The authors reference an article (Cai et al., 2022) that only uses NDVI, and they do not justify if (or why) the NDVI is the most used index. A correct reference that supports this statement is needed..

Response 5: Modified on line 40-41.

Point 6: Line 64-66:“the local climate has also changed significantly due to the impact of global warming”

In what sense? References are needed..

Response 6: Supplemented on line 80-82.

Point 7: line 68:“Therefore, the YREB has complete research elements, which is unique in the world and few scholars have notice that. It is an excellent research area where the changes of NDVI and the driving factors behind them can be studied”

This statement is not important in a scientific article. Of course, each area is “unique” globally, and almost every article could include this. If it is so unique, it should be sustained by several references.

Response 7: Some sentences were deleted and modified on line 82-86.

Point 8: Line 99-105, Authors should cite the datasets properly. Moreover, they should explain the instruments. i.e. AVHRR.

Response 8: The information of the two sets of data is supplemented on line 123-130, line 138-140.

Point 9: Line 149:“cluster analysis […] multielement things”.

“Things” is an inappropriate word in this context. Moreover, the authors should explain this point better.

Response 9: Modified on line 181, and the details are supplemented in line 181-184.

Point 10: Line 174:“correlation between GIMMS and MODIS data is low”

Why the data between the two datasets it is not consistent?.

Response 10: Your comments are very professional. The reasons for the difference between the two are complex, some of which are as follows: There are several aspects of AVHRR sensor design that are not ideal for vegetation trend studies, such as post-launch degradation in sensor calibrations and drift in the satellite overpass times. The seasonal variations in sun-sensor viewing geometry (as defined by the Bidirectional Reflectance Distribution Function, BRDF) combined with sensor drift over time has a large effect on time series of observed NDVI for a given location. Additionally, the spectral configuration of the AVHRR sensors (number of bands, wavelengths covered and the specific band Spectral Response Functions (SRF)) does not permit an accurate atmospheric correction scheme to be applied and absorption and scattering by atmospheric components such as water vapour are a source of error in AVHRR estimates of surface NDVI. The AVHRR channel 2 (near-infrared band) covers wavelengths in which there is considerable absorption by water vapour in the atmosphere, influencing observed NDVI. Therefore, the noise of GIMMS data is large. Moreover, the near-infrared band and the red band difference between GIMMS and MODIS will inevitably lead to the difference of NDVI. It is a complex project to correct the two sets of data, and I can't complete it.

Point 11: Line 176, And what happened in 1997?

The authors should explain the forward and backward curves.

Response 11: Your proposal is interesting, but the changes before and after 1997 are a complex process. Due to the lack of statistical data and affected by the accuracy of GIMMS data, I can't make a more detailed analysis further.

Point 12: Line 235-245, Since these findings are not new (NDVI is well-correlated with temperature and precipitation), authors should link their work with previous scientific literature. It is not noted in the discussion.

Response 12: Modified on line 398-407.

Point 13: Figure 5-6, Why did the authors select five-year smoothed data and not other?

Response 13: Because we want to explore the long-term trend changes, and the policies and activities of the Chinese government are mainly based on five years, which namely five-year plan.That is why we use the five-year moving average.

Point 14: Line 269: “the uptrend was much weaker” Why? It should be explained in the discussion.

Response 14: Revised on line 298-310.

Point 15: Line 344, It should be interesting to show a slope map to justify these statements.

Response 15: Your suggestion is very appropriate. If there is a chart, it must be easy for readers to understand. However, my data has only a few numbers, which is difficult to find and cannot support charts or tables.

Point 16: Line 366, The authors don’t show any data that supports the statements regarding CO2.

Response 16: I added the relevant content on line 430-438.

Point 17: Line 372, Authors should explain the platforms in the material and methods section or introduction.

Response 17: Supplemented at line 131-132 and line177-178.

Point 18: Line 403: “LAI” is not a vegetation index. It can be estimated using vegetation indices, but it’s not one of them. Authors should re-write conclusions.

Response 18: I've put that paragraph in the discussion, on line 408-410.

Thank you for your review and many valuable comments. We look forward to hearing from you regarding our submission. We would be glad to respond to any further questions and comments that you may have. The revised manuscript is in the attachment.

Round 2

Reviewer 1 Report

I understand that other reviewers have uploaded their review and this may
be acceptable. I agree with their remarks overall, but I feel the
authors should address the lack of applying and filtering NDVI data
based on quality control for this paper. If the authors are not going to
filter out low quality data, there should be a clear statement made that
states (1) they did not filter low quality data, (2) why they did not do
so, and (3) what the effects of this might be for their results and
conclusions. 

Author Response

Thank you very much for your comments. The manuscript has been revised. Please see the attachment.

Reviewer 3 Report

Dear author,

thanks for update. I am satisfied with your comments and imrovements

Author Response

Thank you for your recognition of our manuscript.

Reviewer 4 Report

The authors have been greatly improved the article, and they addressed all questions.

This reviewer suggests two little modifications before its publication in order to enrich the paper:

Point 10: Line 174:“correlation between GIMMS and MODIS data is low”

Why the data between the two datasets it is not consistent?.

Response 10: Your comments are very professional. The reasons for the difference between the two are complex, some of which are as follows: There are several aspects of AVHRR sensor design that are not ideal for vegetation trend studies, such as post-launch degradation in sensor calibrations and drift in the satellite overpass times. The seasonal variations in sun-sensor viewing geometry (as defined by the Bidirectional Reflectance Distribution Function, BRDF) combined with sensor drift over time has a large effect on time series of observed NDVI for a given location. Additionally, the spectral configuration of the AVHRR sensors (number of bands, wavelengths covered and the specific band Spectral Response Functions (SRF)) does not permit an accurate atmospheric correction scheme to be applied and absorption and scattering by atmospheric components such as water vapour are a source of error in AVHRR estimates of surface NDVI. The AVHRR channel 2 (near-infrared band) covers wavelengths in which there is considerable absorption by water vapour in the atmosphere, influencing observed NDVI. Therefore, the noise of GIMMS data is large. Moreover, the near-infrared band and the red band difference between GIMMS and MODIS will inevitably lead to the difference of NDVI. It is a complex project to correct the two sets of data, and I can't complete it.

Please, add this information to the discussion. These observations enrich the article and could be important for further studies.

Point 13: Figure 5-6, Why did the authors select five-year smoothed data and not other?

Response 13: Because we want to explore the long-term trend changes, and the policies and activities of the Chinese government are mainly based on five years, which namely five-year plan.That is why we use the five-year moving average.

Please, add this information to the material and methods or results section, since it is relevant for understanding why this parameter has been chosen, and could facilitate the work of other scientists trying to replicate the study in other areas.

Author Response

Thank you for your comments. I agree with your. According to your suggestions, I have made supplements on pages 263-266 and 432-447 respectively.